# Re-Enlightenment of Fulminant Type 1 Diabetes under the COVID-19 Pandemic

**DOI:** 10.3390/biology11111662

**Published:** 2022-11-15

**Authors:** Hiroyuki Sano, Akihisa Imagawa

**Affiliations:** Department of Internal Medicine (I), Osaka Medical and Pharmaceutical University, Takatsuki 569-8686, Japan

**Keywords:** fulminant type 1 diabetes, autoimmune type 1 diabetes, coronavirus disease 2019, severe acute respiratory syndrome coronavirus-2, enterovirus, innate immunity, acquired immunity

## Abstract

**Simple Summary:**

From 2019, coronavirus disease 2019 (COVID-19) caused by the novel severe acute respiratory syndrome coronavirus-2 (SARS-CoV-2) spread worldwide and became a serious problem. It is known that various viruses are involved in the development of type 1 diabetes (T1D), but the reports of an association between SARS-CoV-2 and T1D are mixed. Fulminant type 1 diabetes (FT1D) is a subtype of T1D mainly reported from East Asia. Autoimmune type 1 diabetes (AT1D) is characterized by chronic pancreatic β-cell destruction, whereas FT1D is characterized by rapid and almost complete islet destruction following flu-like symptoms, leading to ketoacidosis and, in some cases, sudden death. When discussing the association between COVID-19 and T1D, it is also necessary to focus on FT1D. However, it is not easy to diagnose this subtype without understanding the concept. Therefore, the authors hereby review the concept and latest findings, hoping that the association between COVID-19 and T1D will be adequately evaluated in the future.

**Abstract:**

Fulminant type 1 diabetes (FT1D) is a subtype of type 1 diabetes (T1D) that is characterized by the rapid progression to diabetic ketoacidosis against the background of rapid and almost complete pancreatic islet destruction. The HbA1c level at FT1D onset remains normal or slightly elevated despite marked hyperglycemia, reflecting the rapid clinical course of the disease, and is an important marker for diagnosis. FT1D often appears following flu-like symptoms, and there are many reports of its onset being linked to viral infections. In addition, disease-susceptibility genes have been identified in FT1D, suggesting the involvement of host factors in disease development. In most cases, islet-related autoantibodies are not detected, and histology of pancreatic tissue reveals macrophage and T cell infiltration of the islets in the early stages of FT1D, suggesting that islet destruction occurs via an immune response different from that occurring in autoimmune type 1 diabetes. From 2019, coronavirus disease 2019 (COVID-19) caused by the novel severe acute respiratory syndrome coronavirus-2 (SARS-CoV-2) spread worldwide and became a serious problem. Reports on the association between SARS-CoV-2 and T1D are mixed, with some suggesting an increase in T1D incidence due to the COVID-19 pandemic. When discussing the association between COVID-19 and T1D, it is also necessary to focus on FT1D. However, it is not easy to diagnose this subtype without understanding the concept. Therefore, authors hereby review the concept and the latest findings of FT1D, hoping that the association between COVID-19 and T1D will be adequately evaluated in the future.

## 1. Introduction

Type 1 diabetes (T1D) is characterized by almost absolute insulin deficiency due to the destruction of pancreatic β-cells. Most cases of diabetes are autoimmune type 1 diabetes (AT1D), which is positive for islet-related autoantibodies, and the period from the onset of diabetes symptoms to diabetic ketoacidosis (DKA) ranges from a few weeks to several months. Another form is idiopathic type 1 diabetes, in which islet-related autoantibodies are not detected. In 2000, another idiopathic subtype called fulminant type 1 diabetes (FT1D) with an extremely rapid course was reported [1]. Autopsy tissue from patients who died immediately after FT1D onset showed macrophages and T cells infiltrating the islets, and in some cases, enteroviruses were detected in the islets [2]. Furthermore, pancreatic biopsies performed 1–5 months after onset showed that the islets had almost disappeared, suggesting that they had been destroyed completely in a short period of time [1]. Typical FT1D cases are characterized by DKA after a few days of flu-like or gastrointestinal symptoms, and some patients die suddenly after being diagnosed with the common cold. Considering the probable association of FT1D with not only enteroviruses but also other viral infections, it is important to monitor the impact of the novel severe acute respiratory syndrome coronavirus-2 (SARS-CoV-2) causing the coronavirus disease 2019 (COVID-19) on the incidence of FT1D and to raise global awareness of FT1D. Therefore, this review summarizes the overview and recent findings of FT1D so as not to overlook COVID-19 associated FT1D.

## 2. Clinical Characteristics

To date, most reported FT1D cases have been from East Asia, particularly Japan. FT1D accounts for approximately 20% of newly diagnosed T1D with ketosis or ketoacidosis in Japan [3], making it an important subtype for investigation. It was reported that FT1D accounted for 7.1% of newly diagnosed T1D patients in South Korea and 1.5–5.5% in China [4,5] In Japan, there is no official registration for T1D, so it is difficult to know the exact number of patients with T1D. In the Ehime study, a regional hospital-based study for diabetes, the prevalence of FT1D was reported to be 8.9% in all T1D patients and 0.2% in consecutive 4980 newly diagnosed diabetes patients in Japan [6]. In 2009, Japan was estimated to have 5000 to 7000 FT1D patients based on the estimated number of diabetes patients at that time, the data from the Ehime study, and the data based on multicenter joint research of FT1D [7]. Recently, from 2016 to 2017, the Japanese Ministry of Health, Labor, and Welfare estimates that there were around ten million diabetes patients and approximately 120,000 T1D patients in Japan. Therefore, authors speculate that there are approximately 10,000 patients with FT1D in Japan now, and the number is increasing year by year [8].

According to the Japan Diabetes Society, AT1D is more common among women, whereas no sex-specific difference was observed in FT1D [3]. More than 90% of FT1D patients have adult onset, and the mean age of onset is higher than that of AT1D [3]. Approximately 72% of FT1D patients develop after flu-like symptoms and rapidly progress to DKA [3]. The duration from initiation to completion of pancreatic β-cell destruction is also speculated to be consistent with this short duration. In contrast, patients with AT1D usually progress to DKA approximately within three months after the onset of diabetic symptoms. Furthermore, pancreatic β-cell destruction gradually progresses over several months before the onset of diabetic symptoms and continues after the onset (Table 1) [3]. FT1D tends to be recognized as acute-onset T1D, which is mostly AT1D, because DKA appears acutely: however, the rapidity of destruction of pancreatic β-cells in both is drastically different.

It has been reported that some patients who happened to develop FT1D during treatment for other diseases showed transient hypoglycemia and relatively high insulin levels shortly before the onset [9,10]. This interesting finding may reflect the rapid destruction of pancreatic β-cells and insulin leakage. At the time of diagnosis, blood and urinary C-peptide levels are already very low, suggesting short-term severe pancreatic β-cell destruction. In addition, glutamic acid decarboxylase antibody (GADAb), which is a representative serum marker used for the diagnosis of AT1D, has not been identified in most FT1D patients. In addition, some patients show elevated levels of exocrine pancreatic enzymes or acute pancreatitis; therefore, in FT1D, both islets and pancreatic acinar cells seem to be destroyed in different manners compared to their destruction in AT1D.

Patients with FT1D often consult a physician for flu-like or gastrointestinal symptoms and then develop DKA a few days later. Alternatively, when patients with FT1D who developed DKA consult a physician, they may be diagnosed with a common cold for the preceding flu-like symptoms. DKA is more severe in FT1D than in AT1D. Therefore, fatality is likely if diagnosis is delayed. On the other hand, some patients with FT1D may have been diagnosed with atypical T1D and not recognized as FT1D. To diagnose this T1D subtype, the Japan Diabetes Society has created criteria for definite FT1D diagnosis (Table 2) [11]. However, immune checkpoint inhibitors (ICIs), recently used as anti-cancer drugs, can cause atypical FT1D (ICI-related FT1D), which is one of the immune-related adverse events (irAEs) [12]. In such cases, caution is required in the use of this diagnostic criteria. Caution is also required in FT1D patients that develop in the course of type 2 diabetes because the HbA1c level at the onset is often high and one of the Japanese FT1D diagnostic criteria, “HbA1c < 8.7,” cannot be applied. Therefore, the identification of biomarkers specific to FT1D is anticipated for more definitive diagnosis. Amylase alpha-2A and heat shock protein 10 (HSP10) autoantibodies that are detected in autoimmune pancreatitis are also commonly identified in FT1D and are being discussed as possible diagnostic markers. [13,14]. The comprehensive antibody analysis of FT1D patient serum found CD300e antibodies in the early stages after onset, suggesting their use as an early diagnostic marker [15]. CD300e is a molecule involved in immunoregulation of the surface markers of monocytes and dendritic cells; the significance of CD300e and its antibody is under investigation.

## 3. Treatment of FT1D

The treatment strategy of insulin therapy for FT1D is the same as that for AT1D. However, insulin secretion in AT1D patients is reduced but maintained for a certain period (even decades) after onset [16,17], whereas patients with FT1D have depleted insulin secretion capacity at diagnosis due to islet destruction [3], which may lead to more unstable glycemic control than AT1D. A 2007 report by the Japan Diabetes Society stated that rates of microvascular complications 5 years after onset were more frequent in FT1D patients than in AT1D patients; this result may reflect glycemic instability and low C-peptide levels [18]. Moreover, when HbA1c levels and systolic blood pressure are managed properly, there is no difference in the frequency of microvascular complications between FT1D and AT1D patients [19]. We expect that the lower risk of complications will be facilitated by better treatment owing to new insulin formulations and devices such as sensor-augmented pumps (SAPs) and closed-loop systems.

The primary and available treatment of T1D is insulin therapy as mentioned above, and in some cases that meet the criteria, islet transplantation, pancreatic transplantation, and simultaneous pancreas and kidney transplantation are performed.

However, the future challenge in T1D is the establishment of immunotherapy that prevents the destruction of pancreatic β-cells. Because more residual pancreatic β-cells are beneficial for glycemic control and prognosis, establishing remission therapies for the early stage of AT1D and prevention therapies for high-risk individuals of AT1D is critical [20]; various trials have been performed. Tertiary prevention for new-onset patients with AT1D aims to preserve the remaining pancreatic β-cells to induce and prolong partial remission. Several immunomodulatory and immunosuppressive agents, such as cyclosporine A, anti-CD3, cytotoxic T-lymphocyte-associated protein 4 (CTLA4)-Ig, anti-CD20, and recombinant human glutamic acid decarboxylase formulated with aluminum hydroxide(rhGAD65-alum), have been investigated to prevent the destructive autoimmune process of pancreatic β-cells that occurs in T1D [21,22,23,24,25,26,27]. Some trials have been successful in partially retaining residual pancreatic β-cell function; however, patients with FT1D are not indicated for tertiary prevention because their pancreatic β-cell function is already depleted at disease onset.

In AT1D, primary prevention trials for individuals with high-risk genotypes in which islet-associated autoantibodies have not yet emerged, and secondary prevention trials for individuals with islet-associated autoantibodies but without diabetes were conducted, some of which are currently ongoing. Of these, their effectiveness has been reported only in limited trials, such as secondary preventive trials using anti-CD3, and the results of other ongoing trials are expected [28]. However, in FT1D, there is no strategy to identify individuals at risk; thus, it is difficult to establish a primary or secondary prevention program.

The most important candidate virus for AT1D is the Coxsackie B virus. Vaccines against the Coxsackie B virus have recently been developed and have been shown to suppress Coxsackie B virus-induced diabetes in mice [29]. If the vaccine were applied to humans, it may be useful for the management of various diseases caused by Coxsackie B virus, including AT1D. 

Since FT1D has also been suggested to be associated with enterovirus, including Coxsackie B, if the vaccine were widely used in humans to prevent Coxsackie B virus-associated diseases, it may also have contributed to the prevention of FT1D. However, the etiology of FT1D has not been fully elucidated and the cost-effectiveness of the vaccine must be considered. Therefore, the vaccine program against FT1D is a future challenge.

## 4. Etiology

### 4.1. Disease-Susceptibility Genes

Investigation using the candidate gene approach showed an association of FT1D with the class II human leukocyte antigen (*HLA*) gene, similar to AT1D. First, it was revealed that a high rate of 41.8% of Japanese FT1D patients possessed the HLA class II haplotype DR4-DQ4 [30]. Genotyping showed that the highest disease sensitivity was at *DRB1*04:05-DQB1*04:01*, followed by *DRB1*09:01-DQB1*03:03*, which are haplotypes that are also implicated in Japanese AT1D. In contrast, other sensitive and resistant haplotypes confirmed in AT1D have not been found to be associated with FT1D; these data indicate that there are similarities and differences in the HLA gene susceptibility between the two subtypes of type 1 diabetes. 

In addition, GADAb were detected in approximately 5% of FT1D patients, in which DRB1*09:01-DQB1*03:03 exhibited sensitivity whereas DRB1*04:05-DQB1*04:01 did not [31]. Thus, the involvement of HLA in FT1D differs between cases that are positive or negative for GADAb. 

Recently, the Japan Diabetes Society conducted a genome-wide association study (GWAS) in Japanese FT1D patients. In addition to showing the strongest association with class II HLA genes, which is already known, FT1D showed an association with the *CSAD/lnc-ITGB7-1* region in the long arm of chromosome 12 [32]. Furthermore, fine mapping identified multiple single nucleotide polymorphisms (SNPs) in this region [32]. These SNPs showed no association with AT1D; this result indicates their association with FT1D-specific pathologies.

*CSAD* encodes cysteine sulfinic acid decarboxylase, a taurine synthase, and taurine is known to have antioxidant and anti-apoptotic effects. In animals, taurine has been reported to exert a cytoprotective action on pancreatic β-cells during their destruction by streptozotocin [33]. Therefore, *CSAD* mutation may weaken the protective action of taurine on pancreatic β-cells. 

Furthermore, a long non-coding RNA, *lnc-ITGB7-1*, which affects the expression of *ITGB-7*, overlaps in the *CSAD* region. *ITGB-7* encodes integrin subunit beta 7, and the integrin β chain is expressed on lymphocytes as a heterodimer with the α chain. Additionally, mucosal addressin cell adhesion molecule-1 (MADCAM-1), which is expressed in areas of inflammation, is a ligand of ITGB-7 and involved in lymphocyte migration to areas of inflammation [34]. Therefore, SNPs in this region may also play a role in islet fragility and inflammation and thereby be associated with FT1D.

### 4.2. Viruses and FT1D

As many FT1D patients develop after flu-like symptoms, viruses are strongly suspected to trigger the onset of this disease. In 2008, the Japan Diabetes Society analyzed paired serum from 38 FT1D patients and found that 7 of the 38 patients had elevated antibody levels for one or more of the following: coxsackievirus, rotavirus, cytomegalovirus (CMV), Epstein-Barr virus (EBV), human herpes virus (HHV)6, or HHV7 [35]. In fact, many cases have been reported wherein enterovirus, coxsackievirus, cytomegalovirus, HHV6, influenza virus B, mumps cold virus, EBV, and rotavirus were involved in the development of FT1D [36,37,38,39,40,41,42,43,44,45], and viral infection is considered to be the most important environmental risk factor for FT1D. 

Although these viruses have a global distribution and infect people worldwide, FT1D mainly occurs in East Asia and in Japan, where the majority of FT1D cases have been reported to date.

A national survey in Japan found enterovirus IgA titers to be significantly higher in FT1D patients than in healthy individuals or AT1D patients and suggested the involvement of enteroviruses in FT1D pathophysiology [46]. This method of detecting enterovirus antibodies reacts with several different serotypes of enterovirus, such as coxsackie A, coxsackie B, and echo virus. Titers of IgA would be increased if different serotypes of enterovirus repeatedly infected a single patient. Thus, this result suggests that FT1D patients are more susceptible to enterovirus infections than others. It is interesting to agree with the report that enterovirus VP1 was detected in the autopsy pancreatic tissue of FT1D. Similar to FT1D, it has been pointed out that enterovirus, CMV [47], rotavirus [48] etc., are involved in AT1D, particularly the contribution of enterovirus is extensively supported [49].

Enteroviruses have been isolated from intestinal biopsy samples in 75% of T1D patients versus 10% of control patients, possibly reflecting persistent enterovirus infection of the gut mucosa [50]. The intestine, which is anatomically close to the pancreas, may contribute as a reservoir of enterovirus in T1D. Alternatively, initial immune recognition could occur in the intestine, followed by subsequent homing of activated lymphocytes into the pancreas [50].

The Diabetes Virus Detection study (DiViD) detected enterovirus in pancreatic tissue specimens from three of six T1D patients [51]. Further, enterovirus proteins, enterovirus RNA, and an active antiviral host response have been demonstrated in the pancreata of T1D donors from biobanks including the Network for Pancreatic Organ Donors with Diabetes (nPOD) [52]. These data suggest that enteroviruses are persistently or repetitively infected during the process of T1D.

The chronic autoimmune process of AT1D is defined by three stages of disease progression [53]. Stage 1 is seroconversion to one or more islet-related autoantibodies [53]. Stage 2 is damage to the pancreatic β-cells causing pre-symptomatic hyperglycemia, and stage 3 is overt T1D due to pancreatic β-cell failure with a requirement for exogenous insulin [53]. The association between enterovirus and AT1D can be summarized as follows. Inefficient viral clearance of enterovirus [54], cytokine response from pancreatic β-cells and increased expression of MHC class I on pancreatic β-cells cause or accelerate islet autoimmunity through molecular mimicry, inflammation, and bystander activation [49]. That is, enteroviruses have the ability to initiate islet autoimmunity and accelerate all three stages. Recently, it was demonstrated that enterovirus isolated from a newly diagnosed T1D patient could infect human islet cells and induce destruction in vitro [55].

From the above, it is considered that in sensitive individuals, enterovirus directly lyses pancreatic β-cells to activate immunity, thereby leading to FT1D. However, further investigation is needed on the significance of repetitive enterovirus infections, the association with other viruses, and the reasons for the rapidity of islet destruction.

### 4.3. Drug-Induced Hypersensitivity Syndrome (DIHS) and FT1D

DIHS is a form of severe drug eruption caused by certain drugs, including carbamazepine, allopurinol, lamotrigine, phenobarbital, and mexiletine. The use of these drugs leads to delayed-onset severe erythema, fever, liver dysfunction, and lymph node swelling, followed by the reactivation of HHV6 after 2–3 weeks, which is characteristic of DIHS and is accompanied by a variety of organ disorders, such as encephalitis, FT1D, and nephropathy [56,57]. 

Treatment for DIHS is systemic steroid administration. However, steroids are not considered a major factor in viral reactivation, as reactivation of HHV6 has been observed even in steroid-naïve patients. At DIHS onset, an increase in the number of regulatory T cells and a decrease in the number of B cells have been observed, along with reduced serum IgG levels, suggesting that the state of transient immunodeficiency initially caused by the drug leads to reactivation of the virus [58]. Sugita et al. also speculated that the accumulation of dendritic cells in DIHS skin lesions reduces the number of dendritic cells in the blood, leading to reactivation of the virus [59]. HHV6 reactivation is followed by reactivation of the EBV, CMV, and varicella zoster virus (VZV) [60]. CMV-infected cells in pancreatic tissues of patients with FT1D that developed 3 weeks after DIHS have also been reported [61]. Additionally, effector T cells are activated against viruses, and various forms of organ damage can persist for long periods, even after viral clearance.

There have been many reports of FT1D following DIHS in Japan. According to a nationwide survey, the frequency of FT1D in DIHS is 0.54%, which is much higher than the rate of Japanese spontaneous occurring rate (0.01%) [6]. In many cases, FT1D onset is reported to occur after the eruption peak and during steroid tapering [62]. Therefore, an excessive immune response, which destroys the islets, is considered to occur due to the reactivation of a virus with an affinity for pancreatic β-cells and the recovery of immune functions by stopping the causative drug or reducing the dose of steroid.

### 4.4. COVID-19 and FT1D

Previous studies have shown that SARS-CoV-2 infects pancreatic endocrine and acinar cells via angiotensin-converting enzyme 2 and neurophilin-1 (NRP1), causing pancreatic β-cell dysfunction [63,64]. In fact, viral replication and apoptosis have been observed in pancreatic β-cells infected with SARS-CoV-2 [64,65,66,67]. Alternatively, upon SARS-CoV-2 infection, pancreatic β-cells show low insulin expression, high glucagon expression, and high trypsin 1 expression, suggesting transdifferentiating of pancreatic β-cells [64,68]. These facts suggest the possibility of pancreatic β-cell dysfunction caused by SARS-CoV-2 leading to virus-induced diabetes [68]. If an individual before the onset of AT1D or type 2 diabetes suffers from COVID-19, pancreatic β -cell damage may accelerate the onset of diabetes. This assumption is consistent with the results of epidemiological studies showing an increased risk of developing diabetes mellitus in those with COVID-19 [69]. 

Some patients who develop diabetes in the process of COVID-19 also improve their diabetes with the improvement of COVID-19 [70]. In such cases, inflammation, cytokines, antivirals, and steroid therapy are considered to have caused hyperglycemia, in addition to transient viral cell damage [71,72,73,74,75]. 

Interestingly, SARS-CoV-2 has also been shown to infect macrophages via ACE or CD16. Infection of macrophages by SARS-CoV-2 activates inflammasomes and drives pyroptosis. Pyroptosis interrupts the viral replication cycle but releases immune cell activators and triggers a hyper-inflammatory state by other macrophages [76]. This hyper-inflammatory state is supposed to be involved in the exacerbation of COVID-19, but it is unknown whether it is involved in the development of diabetes. Importantly, there is a lack of evidence that SARS-CoV-2 infection initiates immune disorders leading to AT1D or FT1D.

There are mixed epidemiological reports of the association between COVID-19 and the development of T1D. Although some regions have reported increased incidences of T1D after the spread of COVID-19, others did not report this association; therefore, there is a lack of consensus in this regard [77,78]. Recent real-world data indicates COVID-19 diagnosis is associated with increased risk of new-onset T1D in American Indian and Alaskan Native peoples, Asian and Pacific Islander peoples, Black populations, infants, and the elderly [79].

In areas with strict lockdown and other infection control measures, there has been a decrease in the incidence of common viral infections, such as enterovirus infections, which may even lead to fewer T1D patients in the future [78]. Therefore, this trend may vary depending on regional characteristics, race, and age, and requires further investigation. 

During the pandemic, the frequency and severity of DKA in newly diagnosed T1D patients were higher than before [80,81,82]. It is important to investigate whether these severe cases mean FT1D or not. However, according to the details of many case reports, most cases have not met the Japanese diagnostic criteria for FT1D because of the high value of HbA1c or the individual is positive for GADAb. Most of the newly diagnosed T1D patients in these epidemiological studies were negative for SARS-CoV-2 at the onset [83,84]. Furthermore, no case of FT1D or increase in FT1D incidence has been confirmed, even in Japan. Therefore, as these epidemiological studies conclude, delays in medical access attributable to the lockdown or fear of infection may have caused delays in the treatment of new-onset AT1D, leading to DKA [85,86]. Currently, there is little evidence of a causal link between SARS-CoV-2 and FT1D, but during this pandemic, it is important to pay attention to the rapidity of the diabetic clinical course in patients with relatively low HbA1c levels at diagnosis so as not to overlook FT1D.

To protect from the severe pandemic of COVID-19, various anti-COVID-19 vaccines have been developed [87] and approximately 60% of the world’s population has received at least one dose of vaccination (https://ourworldindata.org/covid-vaccinations Coronavirus (COVID-19) Vaccinations—Our World in Data, accessed on 2 November 2022). 

Recently, case reports of FT1D occurring after COVID-19 vaccination are arriving from mainly East Asia. At this time, what these reports have in common is that the patients had risk factors for FT1D, such as disease susceptible HLA alleles or a past history of use of immune checkpoint inhibitors [88,89,90]. Therefore, these reports caution clinicians that FT1D might develop after COVID-19 vaccination in individuals with susceptible backgrounds. 

The long history of vaccines has shown that the immune-stimulatory properties of vaccines can trigger immune disorders including autoimmune diseases [91,92]. In addition, molecular mimicry between the SARS-CoV-2 spike (S) protein and human endocrine cells, including pancreatic β-cells, has been pointed out. In fact, both widely used mRNA and adenovirus vector-based vaccines encode the S protein, which is targeted by neutralizing antibodies [87,93]. Therefore, we cannot exclude the possibility that vaccine-induced immune activation and molecular mimicry might be involved in the development of FT1D. On the one hand, it is also possible that the timing of vaccination and FT1D onset coincided by chance. Therefore, accumulation of further cases is necessary to clarify the causal relationship between the vaccination and the development of T1D. 

### 4.5. Mouse Diabetes Model Induced by Encephalomyocarditis Virus (EMCV)

Like enterovirus, EMCV belongs to the Picornaviridae family. EMCV is the most thoroughly studied diabetes-associated virus in mice and much evidence has been accumulated via a mouse model that rapidly develops diabetes [94]. Since the concept of human FT1D was reported, attention has been focused on the similarities between this model and FT1D [95].

First, it was reported that the injected EMC M variant (EMC-M) virus induces diabetes as a result of pancreatic β-cell infection, but the onset was inconsistent. Therefore, EMC-M viral plaque purification was performed to isolate two mutants: diabetes-induced EMC-D virus and a non-diabetes EMC-B virus [96]. 

Although EMC-D differs from the B variant (EMC-B) by only a few nucleotide bases, EMC-B does not cause diabetes, suggesting that naturally occurring genetic mutations in a virus can alter its capacity to induce diabetes. Sensitivity to EMCV is affected by age, gender, and genetic factors. As a genetic factor, susceptibility varies depending on the mouse strain: SJL/J and DBA/2 are sensitive, while C57B/6 is resistant [96]. 

In sensitive mice, EMC infection causes severe inflammatory cell infiltration into islets, and the inflammation extends to the exocrine pancreatic region. The inflammatory cells mainly comprise of macrophages, and the peak timepoint is 72 to 120 h after infection [97]. Severe pancreatic β-cell lysis is caused by extreme “high-dose” virus injections, which are unlikely to occur in nature [98].

Simultaneously, there is a transient increase in insulin levels and a decrease in blood glucose levels, which are signs of rapid pancreatic β-cell destruction, followed by the rapid disappearance of insulin and the appearance of hyperglycemia [95]. The inflammation disappears within a few days, but the pancreatic β-cells are destroyed, making it difficult for the mice to survive without insulin. This time course is similar to that of FT1D. 

Sensitive SJL/J mice are known to have mutations in the tyrosine kinase 2 (*Tyk2*) gene associated with the capacity of interferon-dependent defense, an important part of innate immunity against viral infections [99]. Introducing this mutation into disease-resistant C57B/6 mice leads to the development of diabetes by EMCV, suggesting that the *Tyk2* mutation is associated with disease sensitivity in mice. Partial dysfunction of innate immunity due to this mutation could lead to an increased viral load [99]. Activating innate immunity with the administration of *Corynebacterium parvum* before EMCV infection can halt the onset of diabetes, demonstrating the importance of early antiviral action in preventing diabetes development [100]. Experiments with T cell or B cell-deficient mice have shown no protective or promoting effect on diabetes, but macrophage inactivation prior to viral infection prevents diabetes.

Upon low-dose infection of mice, the virus initially replicates in pancreatic β-cells, leading to the recruitment of macrophages that are activated at the pancreatic islets. These activated macrophages produce interleukin-1 β (IL-1 β), tumor necrosis factor-α (TNF-α), and nitric oxide (NO), which induces the destruction of pancreatic β-cells [101]. These results indicate that viral pancreatic β-cell lysis and innate immune disorders, such as reduced initial viral clearance and overactivation of macrophages, play major roles in the pathophysiology of fulminant diabetes in the mouse model. 

### 4.6. Pregnancy-Related FT1D (PF)

Reportedly, FT1D tends to occur from the third trimester of pregnancy to immediately after delivery [102]. In women of reproductive age (13–49 years old), FT1D that is unrelated to pregnancy is distinguished from PF as non-pregnancy-related FT1D (NPF). 

In a nationwide survey by the Japan Diabetes Society, PF accounted for 21% of FT1D patients in women aged 13–49 years, and most of the patients who developed T1D during pregnancy had FT1D [102]. The third trimester of pregnancy overlaps with a period of increased insulin resistance, and thus, the severity of DKA is higher than that in NPF, along with a stillbirth or miscarriage rate of 63.6% [102]. 

Despite immune tolerance during pregnancy, PF can develop not only after delivery but also during pregnancy; thus, the similarities and differences between FT1D before and after delivery need to be studied. 

HLA susceptibility also differs between PF and NPF, with *DRB*09:01-DQB1*03:03* (DR9) haplotypes observed more frequently in PF than in NPF [103] and *DRB1*04:05-DQB1*04:01*, the most common haplotype in FT1D, observed more frequently in NPF than in PF of Japanese patients [103]. There is no difference in the frequency of flu-like symptoms between PF and NPF (both about 70%) [103]. Thus, the involvement of viral infections is suspected in both cases; however, this phenomenon has only been verified in a small number of cases and requires further validation. 

### 4.7. Immune Checkpoint Inhibitor (ICI)-Related FT1D

Recently, ICIs such as anti-programmed cell death protein 1 (PD-1) antibodies, anti- programmed cell death ligand 1 (PD-L1) antibodies, and anti-CTLA-4 antibodies have contributed greatly to cancer therapy and have improved the prognosis of advanced cancer patients. In addition, immune checkpoints are crucial for self-tolerance and immune-related adverse events (irAEs) induced by ICIs are concerning. In irAEs, endocrine disorders, enteritis, hepatitis, and interstitial pneumonia are relatively common, and each frequency is over 5–10%, which is at a warning level [12]. T1D is also one of the irAEs and the frequency is rare at approximately 0.1–2.2% [104,105,106]. However, precaution is necessary because ICI-related T1D patients require permanent insulin therapy to protect their lives, and certain ICI-related T1D are fatal if treatment is not initiated shortly after onset [105]. There is no exception in Caucasians, where the FT1D was considered rare. In fact, with the widespread use of ICIs, reports of ICI-related FT1D in Caucasians are increasing [104,106,107].

In recent years, cases of ICI-related diabetes have been reported from various countries, and analysis of these reports is gradually revealing the entity of ICI-related diabetes. As an overview based on some reviews, the average age of onset of ICI-related diabetes is in the 60s, consistent with the cancer-prone age, which is higher than that of classic T1D [106,107,108,109]. Males account for 50–60% patients, slightly more than females, reflecting that the populations treated with ICIs are most commonly used for melanoma and non-small cell lung cancer [104,106,107,108,109]. The onset varies from 13 to 504 days after the initial dose of ICI, sometimes even after the first dose, up to more than one year after initiation of ICI, but the mean or median is four months or four cycles later [104,106,107,108,109,110,111]. 

DKA is a common presentation for ICI-related T1D, with the incidence varying from 39–68% based on some reviews [108]. According to a European report, the frequency of DKA in T1D is 21%, and conventional FT1D usually develops DKA. Therefore, its frequency is intermediate between classic T1D and conventional FT1D [112].

The C-peptide level of ICI-related T1D is usually low or undetectable, but it has also been reported that sometimes C-peptide levels recover slightly after the end of ICI therapy [104]. This is an intermediate feature between classic T1D, in which C-peptide is detected for a relatively longer period of time after onset, and conventional FT1D, in which C-peptide is depleted at the onset. The average or median blood glucose level at the diagnosis of ICI-related T1D is 560–650 mg/dl, which is higher than that of 434 mg/dl in Japanese acute-onset T1D and 385 mg/dl in Caucasian acute-onset T1D, and lower than 853 mg/dl of Japanese FT1D [104]. These facts suggest that the rapidity of pancreatic β-cell destruction is faster than acute-onset T1D and slower than conventional FT1D. The mean HbA1c at the onset of ICA-related diabetes is 7.9% [106], which is lower than that of T1D, approximately 9% [3,113], and this relatively low HbA1c level also suggests faster progression than acute onset type 1 diabetes and resembles conventional FT1D.

In contrast to patients with acute-onset T1D, of whom more than 80% have islet-related autoantibodies, the positive rate in ICI related T1D patients is only approximately 50% [108,114]. Thus, relatively numerous patients belong to idiopathic T1D, clearly distinguishing it from AT1D. Especially, the positive patients are rare in Japanese people [104], so it is expected to vary by race [106,107]. In patients wherein the HLA haplotype was investigated, 51.3% showed the DR4 and 14.1% the DR9 phenotype, while in 10.3%, protective phenotypes for T1D (DR7, DR11, or DR15) were detected [108]. The median blood glucose level at the onset in patients with HLA-DR4 was higher than in non-DR4 HLA patients, and the onset of patients with the T1D protective HLA-DR was later than the others [108]. 

At the onset of FT1D, the elevation of pancreatic exocrine enzyme has been reported in most patients, and some patients were accompanied with pancreatic swelling or acute pancreatitis, supporting the possibility of rapid islet and exocrine injury [115]. Moreover, ICI-related T1D varies from FT1D-like with pancreatitis to classical T1D-like with mild or no pancreatic enzyme elevation. In addition, if elevated pancreatic enzymes are observed at the time of the diagnosis of classic T1D, it may be a nonspecific change based on DKA [116]. In a recent meta-analysis, 51% of patients diagnosed with ICI related T1D had an increase in either lipase and/or amylase at diagnosis, and the frequency is disproportionately higher than that of classic T1D [108]. 

Recently, the concept of AT1D as a combined endocrine–exocrine disease has become commonplace because it has become clear that inflammatory cell infiltration, fibrosis, and acinar cell depletion in exocrine glandular tissue are simultaneously advanced [117]. A decrease in insulin action on acinar cells is considered to be one of the factors of exocrine atrophy, but in preclinical ATID and even in slowly progressive type 1 insulin-dependent diabetes mellitus (SPIDDM)/ latent autoimmune diabetes in adults (LADA) whose insulin secretion is relatively maintained, morphological changes in the exocrine glands have been observed. Therefore, it is assumed that important immune mechanisms that affect exocrine tissue exist in T1D, and it is also necessary to investigate whether this mechanism, and that of pancreatitis in FT1D and ICI-related T1D, are on the same spectrum.

From these facts, it can be stated that ICI-related T1D has various phenotypes, from FT1D to AT1D. In fact, it has been reported that approximately 40–50% of ICI-related diabetes meet the Japanese diagnostic criteria for FT1D [104]. So, it is also important to focus on the difference between conventional FT1D and ICI-related FT1D. Unlike conventional FT1D, flu-like symptoms are rare in ICI-related diabetes and are rarely reported to be associated with viral infection. Therefore, the direct association between ICI-related FT1D and viruses may be less than that of conventional FT1D.

ICI-related diabetes was primarily caused by PD-1 or PDL1 antibodies and was rarely caused by CTLA4 antibodies. The role of immune checkpoints in T1D has been investigated in mice and humans. Non-obese diabetic (NOD) mice rapidly develop diabetes following the experimental blockade of PD-1 or PD-L1 [113,118]. This corresponds with the finding that pancreatic islets express PD-L1 at low levels in mice and are more upregulated in inflamed islets [119,120]. In humans, polymorphisms in the PD-1 gene confer increased susceptibility to a variety of autoimmune disorders, including T1D [121,122,123]. Therefore, viral infection is considered to trigger conventional fulminant type 1 diabetes, while inhibition of PD-1 or PD-L1 is thought to trigger ICI-related FT1D. 

Based on the above evidence, ICIs are triggers of FT1D in addition to viruses, but it may be appropriate to understand ICI-related diabetes, including ICI-related FT1D, as an independent and a novel category from an etiological approach [108] (Figure 1).

## 5. Immune Mechanism

### 5.1. Innate Immunity and FT1D

The characteristic clinical course of FT1D, that is, onset following flu-like symptoms and rapidity, suggests that innate immunity plays an important role in disease pathogenesis. The innate immune response to viruses involves the natural killer (NK) cell-mediated clearance of virus-infected cells, phagocytosis by macrophages, and suppression of viral proliferation by interferons (IFNs) produced by inflammatory and infected cells. Of the molecules that recognize pathogens, important pattern recognition receptors with a lower level of specificity include macrophage toll-like receptors (TLRs) and the intracellular viral receptors, namely retinoic acid-inducible protein-1 (RIG-I) and melanoma differentiation-associated protein-5 (MDA-5). 

The immunopathology of FT1D is challenging to study in humans because of the extremely rapid clinical course, but some reports revealed the details using pancreas samples removed from patients who died shortly after the onset of FT1D. Tanaka et al. studied autopsied pancreata from three patients and reported prominent infiltration of macrophages, cytotoxic T cells, and dendritic cells into the islets and exocrine region, along with the presence of enterovirus VP1 antigens in the pancreatic islets and exocrine region [124]. Furthermore, Aida et al. reported increased expression of the intracellular viral receptors RIG-I and MDA-5 in islet cells and increased expression of IFN-α/β, IFN-γ, CXCL-10, and FAS in pancreatic β-cells [125]. 

Our analysis of autopsied pancreatic tissue from three patients that died immediately after FT1D onset revealed infiltration of macrophages in 96.2% and T cells in 68.8% of the islets and TLR-3, -7, and -9 expression in most islet cells. Additionally, the study revealed depleted pancreatic β- and α-cells [2]. Furthermore, in situ hybridization confirmed the presence of enteroviral RNA in islets with remnant pancreatic β-cells in one of the three patients [2]. Nishida et al. carried out proteomic analysis of islets in autopsy samples obtained from patients with FT1D and detected new proteins associated with cell migration (lymphocyte cytosolic protein 1), virus replication (adenosine triphosphate-dependent RNA helicase, DEAD box helicase 5), and antiviral activity (SAM domain and HD domain-containing protein 1) [126]. These findings support the hypothesis that viral infection of pancreatic islet cells could induce infiltration of immune cells to the islets, resulting in pancreatic β-cell destruction through immune reactions to the virus, but there is insufficient evidence to explain the rapid and complete mechanism underlying pancreatic β-cell destruction. 

Experiments in EMCV-resistant C57/B6 mice demonstrated that a deficiency of MDA-5 or TLR-3 leads to the development of diabetes, suggesting that attenuated activity of these molecules, especially TLR3, may lead to blunted type1 IFN response, uncontrolled virus replication, and diabetes during pancreatic β-cell tropic virus infection [127].

Some studies suggest that NK cell dysfunction is involved in impaired innate immunity in FT1D patients. Analysis of peripheral blood from FT1D patients revealed that the expression of killer cell lectin-like receptor subfamily C3 (KLRC3) and CD94 that form activated receptors on NK cells, as well as the proportion of NK cells in the peripheral blood mononuclear cells, was lower in FT1D than in healthy subjects [128]. In a similar report, the genetic analysis of peripheral blood mononuclear cells in FT1D patients showed upregulation of interleukin-1 accessory receptor protein (IL1RAP) and downregulation of TLR-9 and the transcription factor ELF4, suggesting a decline in NK cell function. In fact, the NK activity measured by Methyl thiazoleterazolium (MTT) assay was reduced in FT1D patients. [129]. These results provide a hypothesis that the failure of NK cell activation leads to the loss of the ability to eliminate the causative virus. This failure of initial virus elimination is an interesting phenomenon that was also confirmed in EMC model mice; and should be noteworthy in the investigation of FT1D pathogenesis. However, it is unclear whether this phenomenon leads to excessive immune responses by macrophages or cytotoxic T cells.

### 5.2. Acquired Immunity and FT1D

While aberrant innate immunity is considered to be the primary etiology of FT1D, the rapid and particularly complete depletion of pancreatic β-cells suggests the involvement of other factors such as acquired immunity. In addition, the fact that *DRB1*04:05-DQB1*04:01* and *DRB1*09:01-DQB1*03:03* were associated not only with AT1D but also FT1D in Japan raises debate about the involvement of autoimmunity in FT1D.

Secreted CXCL10 activates and recruits not only macrophages but also T cells via CXCR3. This common phenomenon has also been confirmed in AT1D [130]. Overexpression of the interferon-induced chemokine CXCL10 by enteroviral infection of pancreatic islet cells has been proposed as an early molecular marker of infection [131]. In FT1D, the serum levels of CXCL10 were increased at the onset [132]; furthermore, the expression of CXCL10 in pancreatic islets and the accumulation of CXCR3-positive T cells have been confirmed in the autopsy pancreata of three patients [133]. Therefore, it is hypothesized that antiviral and/or autoreactive T cells are also involved in the destruction of pancreatic β-cells.

As mentioned previously, GADAb were detected in approximately 5% of FT1D patients, whereas either islet-related autoantibodies were detected in approximately 18% of FT1D patients [134]. Some FT1D patients negative for GADAb at onset have been observed to become positive over the course of the disease [135]. In addition, T cells reacting with islet components [136], amylase α2 antibodies, and HSP10 antibodies have been detected in the peripheral blood of FT1D patients [13,14], suggesting the involvement of autoimmunity. Tada et al. created CD28-knockout non-obese diabetic (NOD) mice with significantly fewer regulatory T cells and found that insulitis occurred earlier in CD28-knockout than in ordinary NOD mice. They also reported that the administration of poly (I:C) in the mice simulated a viral infection and immediately led to islet destruction, similar to its occurrence in FT1D [137]. In humans, an analysis of the expression of CTLA4 in the peripheral blood T cells of FT1D patients revealed reduced expression of both regulatory T cells and effector T cells, as well as reduced suppression of the proliferation of effector T cells [138,139]. The reduced inhibitory functions of regulatory T cells indicate that FT1D is characterized by an excessive immune response by cytotoxic T cells to virus-infected islet cells. 

These findings indicate that the development of FT1D is associated with aberrant innate immunity as well as impaired regulatory T cell function, which is considered to trigger an exaggerated adaptive immune response against virus and self-antigens, leading to rapid and complete destruction of pancreatic β-cells (Figure 2). 

## 6. Conclusions

In this review, we have outlined the features of FT1D and the role of viruses in FT1D pathophysiology. Diagnosing FT1D without delay is essential to reduce the risk of sudden death.

Patients with FT1D sometimes visit the hospital complaining of severe fatigue following the symptoms of the common cold. If clinicians encounter such cases, it is necessary to ask in detail whether the patient has symptoms of sudden hyperglycemia such as thirst, polydipsia, and polyuria. Additionally, in such cases, urinalysis will be of great help. If urinary sugar and urinary ketone bodies are detected, FT1D is strongly suspected, and blood glucose levels should be measured immediately. 

When conducting an epidemiological study of acute-onset T1D associated with COVID-19, HbA1c at the onset should be focused on. If the HbA1c is less than 8.7%, there is a possibility of FT1D. In addition, the rapid clinical course and low C-peptide levels strongly support the existence of FT1D.

Currently, there is little evidence of an increase in the number of cases of FT1D after SARS-CoV-2 infection. However, it is possible that FT1D was hidden behind the sudden death after COVID-19. Alternatively, FT1D may have been included in diabetes that develops during the course of COVID-19. Therefore, it is necessary to raise global awareness of FT1D and carefully monitor the relationship between FT1D and COVID-19.

Finally, it is also necessary to pay attention to how the temporal high-hygiene environment, owing to measures against COVID-19, will affect T1D and viral infections in the future.

## Figures and Tables

**Figure 1 biology-11-01662-f001:**
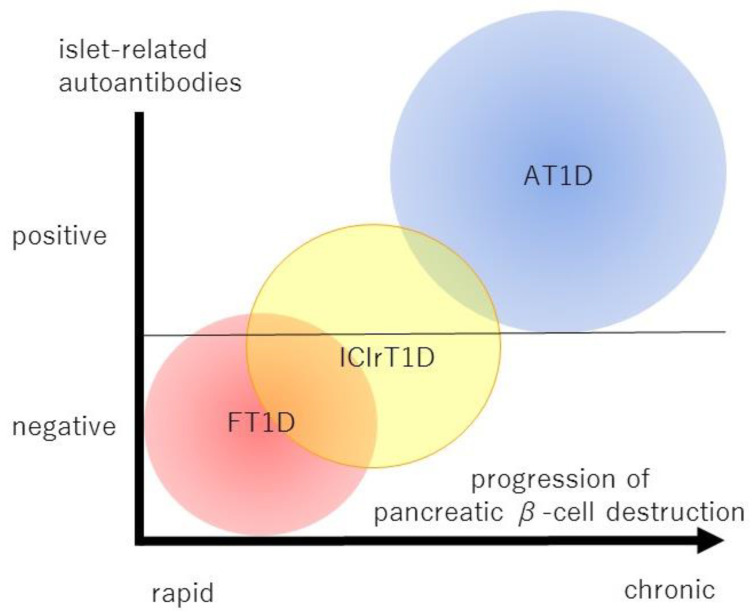
This schema shows the positioning of fulminant type 1 diabetes (FT1D), ICI-related type 1 diabetes (ICIrT1D) and autoimmune type 1 diabetes (AT1D) based on the presence of islet-related autoantibodies and the rapidity of pancreatic β-cell destruction. ICIrT1D exhibits various phenotypes ranging from FT1D-like to AT1D-like, but the average phenotype is intermediate between FT1D and AT1D, and is considered an independent entity from the etiological approach.

**Figure 2 biology-11-01662-f002:**
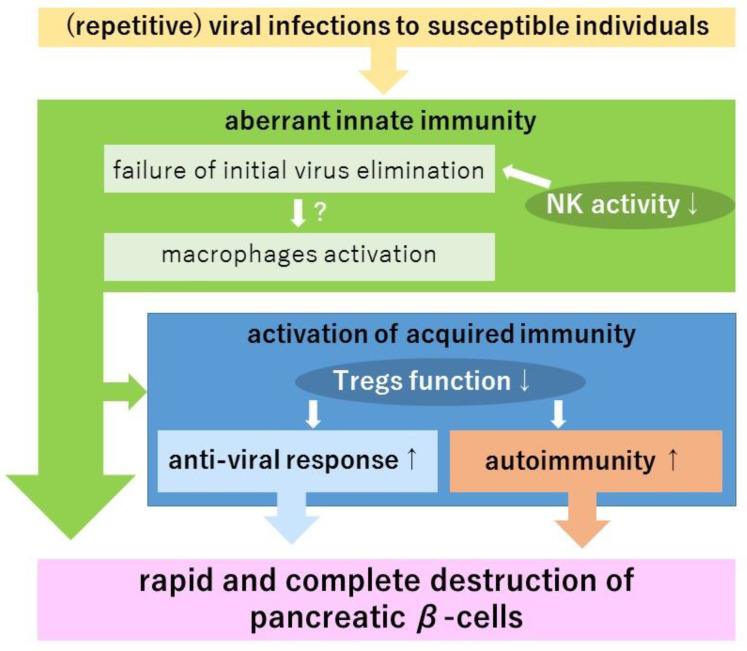
Aberrant immune response leading to pancreatic β-cell destruction in fulminant type1 diabetes (FT1D) (hypothesis). The development of FT1D following flu-like symptoms suggests the importance of aberrant innate immune responses. Decreased natural killer cell (NK) activity has been reported in FT1D patients, suggesting failure of initial virus elimination. Histopathologically, the marked infiltration of macrophages into pancreatic islets is observed. In mouse models, macrophages are considered to be the major factor in pancreatic β-cell destruction. Furthermore, T cell infiltration into pancreatic islets was observed in histopathological tissue. Repetitive enterovirus infections in FT1D patients suggests the existence of acquired immunity against virus-infected cells. In addition, the fact that islet-associated autoantibodies were positive in some cases suggests the existence of autoimmunity. The decreased regulatory T cell (Treg) activity shown in FT1D patients, and this aberrant innate immune response may make antiviral immunity and autoimmunity more active. Although these characteristic immune profiles are hypothesized to contribute to the rapid and complete destruction of pancreatic β-cells, further investigation is required to make this figure more detailed.

**Table 1 biology-11-01662-t001:** Comparison of fulminant type 1 diabetes and autoimmune type 1 diabetes mellitus (created by authors based on [3]).

	FT1D	AT1D
Age at onset	Wide age range but mainly adult	Wide age range but the peak is in adolescence
Sex	Male ≒ female	Male < female
Onset (duration from onset to DKA)	Very rapid (within approximately 7 days)	Acute (within approximately 3 months)
HbA1c level at onset	~8.7%	Very high
Insulin secretion	Depleted at onset	Slight residual for some time after onset
Flu-like symptoms	71.7%	26.9%
Elevation of pancreatic enzyme levels	Frequently	Rarely
Anti-GAD antibody	Mostly negative	Positive
Susceptible HLA haplotype in Japanese	DRB1*04:05-DQB1*04:01 DRB1*09:01-DQB1*03:03	DRB1*09:01-DQB1*03:03 DRB1*04:05-DQB1*04:01 DRB1*08:02-DQB1*03:02

**Table 2 biology-11-01662-t002:** Japanese criteria for definite diagnosis of fulminant type 1 diabetes mellitus (2012) [11].

Fulminant type 1 diabetes mellitus is confirmed when the following three findings are present:
(1) Occurrence of diabetic ketosis or ketoacidosis soon (approximately 7 days) after the onset of hyperglycemic symptoms (elevation of urinary and/or serum ketone bodies at first visit)
(2) Plasma glucose level ≥ 16.0 mmol/L (≥288 mg/dL) and glycated hemoglobin level < 8.7% (NGSP value) * at first visit
(3) Urinary C-peptide excretion <10 μg/day or fasting serum C-peptide level <0.3 ng/mL (<0.10 nmol/L) and <0.5 ng/mL (<0.17 nmol/L) after intravenous glucagon (or after meal) load at onset
Other findings in fulminant type 1 diabetes mellitus:
(A) Islet-related autoantibodies, such as antibodies to glutamic acid decarboxylase, islet-associated antigen 2, and insulin, are undetectable in general.
(B) Duration of the disease before the start of insulin treatment can be 1 to 2 weeks.
(C) Elevation of serum pancreatic enzyme levels (amylase, lipase, or elastase-1) is observed in 98% of the patients.
(D) Flu-like symptoms (fever and upper respiratory symptoms, etc.) or gastrointestinal symptoms (upper abdominal pain, nausea, and/or vomiting, etc.) precede the disease onset in 70% of patients.
(E) The disease can occur during pregnancy or immediately after delivery.
(F) Association with HLA *DRB1*04:05-DQB1*04:01* is reported.

* This value is not applicable for patients with previously diagnosed glucose intolerance. NGSP, National Glycohemoglobin Standardization Program.

## Data Availability

Not applicable.

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
