# Peer review of "Re-Enlightenment of Fulminant Type 1 Diabetes under the COVID-19 Pandemic"

_biology, 2022, doi:10.3390/biology11111662_

Round 1
Reviewer 1 Report (Previous Reviewer 1)
The manuscript including the figures has greatly improved. Thank you for the revisions.
Author Response
Dear reviewer,
Your meaningful suggestions have changed our manuscript for the better. Thank you.
I have proofread one part of the manuscript so that the reader can read it more easily.
- Clinical Characteristics
To date, most reported FT1D cases have been from East Asia, particularly Japan. FT1D accounts for approximately 20% of acute-onset T1D cases in Japan [3], making it an important subtype for investigation.
I changed “acute-onset T1D cases” to “newly diagnosed T1D with ketosis or ketoacidosis”.
We believe this change will make it easier for readers to understand. We appreciate your understanding.
Dear Reviewer1
Thank you very much for your support. Your suggestions for this review are very good and have greatly improved the quality of this review. Thank you very much for your contribution.
Hiroyuki Sano

Reviewer 2 Report (New Reviewer)
This review focusing on fulminant type 1 diabetes under COVID-19 pandemicis well written. Thus, I have no specific comments in this review manuscripts.
Author Response
Dear reviewer,
Your meaningful suggestions have changed our manuscript for the better. Thank you.
I have proofread one part of the manuscript so that the reader can read it more easily.
- Clinical Characteristics
To date, most reported FT1D cases have been from East Asia, particularly Japan. FT1D accounts for approximately 20% of acute-onset T1D cases in Japan [3], making it an important subtype for investigation.
I changed “acute-onset T1D cases” to “newly diagnosed T1D with ketosis or ketoacidosis”.
We believe this change will make it easier for readers to understand. We appreciate your understanding.
Dear reviewer 2
Thank you very much for your support.
It was an honor to have you review it.
Thank you for taking time out of your busy schedule to read this review.
Hiroyuki Sano

Reviewer 3 Report (New Reviewer)
This is a timely and much needed review. I suggest major revision in order to take care of the following points:
- The abstract contains mainly the background, the authors need to summarize what they have done in this review and include it in the abstract.
- Lines 57-69, try to include data from East Asia, other than those from Japan.
- Line 87-88= In addition, glutamic acid decarboxylase antibody (GADAb) has not been identified in most FT1D cases, indicating what? explain why GADAb
- There are reports relating Vaccination against COVID-19 with onset of FT1D. The authors should cite these and discuss the cause of the onset, could it be related to the S protein, in the case of mRNA vaccines?
- The following publication need to be discussed and cited Shirakawa J. Pancreatic β-cell fate in subjects with Covid-19. J Diabetes Investig. 2021;12:2126–2128. doi: 10.1111/jdi.13671
- Clarify more the possible role of islet autoantibodies, genetic background (HLA), and their relationships in the development of FT1D.
- Several data demonstrated increased ACE2 expression in pancreas. This may favor increased pancreatic cells binding of SARS-CoV-2 and hence islet damage. Please discuss this idea and how it may lead to FT1D in COVID-19. Discuss the possible direct infection of the pancrease by the virus (use these references SARS-CoV-2 infects human pancreatic β cells and elicits β cell impairment. Cell metabolism, 2021: p. S1550-4131(21)00230-8. pmid:34081912 and Tang X., et al., SARS-CoV-2 infection induces beta cell transdifferentiation. Cell metabolism, 2021: p. S1550-4131(21)00232-1. pmid:34081913)
- Discuss any possible association between race/ ethnicity and COVID-19 induced FT1D. Association between COVID-19 and T1D differ by race, with the largest risks being among American Indian/Alaskan Natives followed by Asian/Pacific Islanders, Black, Hispanic, and White patients. This may reflect differential risk trajectories of T1D by race/ethnicity that preexisted even before the pandemic. For example, one study found that between 2011 and 2015, the incidence of T1D increased among NH Black patients by 4% per year (95% CI: 1.7, 6.3), Hispanics by 2.5% per year 95% CI: 0.5, 4.6), and Asian/Pacific Islander patients by 8.5% per year (95% CI: 3.2, 14.0), while NH White patients did not see significant annual increases (0.5, 95% CI: -0.7, 1.7) (Trends in incidence of type 1 and type 2 diabetes among youths—selected counties and Indian reservations, United States, 2002–2015. Morbidity and Mortality Weekly Report, 2020. 69(6): p. 161. pmid:32053581 ) Even though the reasons for these differences by race/ethnicity remain unclear. Also, discuss age. Increased risk of COVID-19-related T1D among older adults, while T1D usually affects the younger population (The associations between COVID-19 diagnosis, type 1 diabetes, and the risk of diabetic ketoacidosis: A nationwide cohort from the US using the Cerner Real-World Data https://doi.org/10.1371/journal.pone.0266809)
- Proof read and check the language of the manuscript. For example:
Line 80 spell out DKA= diabetic ketoacidosis (DKA)
Line 81 add “is”= in both is drastically different
Line 106= replace which by and
Line 108, replace its by their
Line 139= remove ‘the’ and add ‘disease’= at disease onset
Author Response
Dear Reviewer3
It was an honor to have your review. Thank you for taking time out of your busy schedule to
read this review. I’ve edited my manuscript according to your pertinent indications.
Answers to each question are written in red. Corrections also are indicated in red in the
proofread manuscript.
The abstract contains mainly the background, the authors need to summarize what they have done in this review and include it in the abstract.→ Your point was appropriate.I added sentences of simple summary to the abstract to make it even better.
- Lines 57-69, try to include data from East Asia, other than those from Japan.
→ I added frequency of fulminant type 1 diabetes in China and South Korea to the paragraph 2. Clinical Characteristics.
- Line 87-88= In addition, glutamic acid decarboxylase antibody (GADAb) has not been identified in most FT1D cases, indicating what? explain why GADAb
→ I added to this sentence that “the GAD antibody is a representative serum marker for diagnosing AT1D.”. Your suggestion made the following sentence which means "FT1D and AT1D have different etiologies“ more understandable.
- There are reports relating Vaccination against COVID-19 with onset of FT1D. The authors should cite these and discuss the cause of the onset, could it be related to the S protein, in the case of mRNA vaccines?
→Thank you, this is a very important point. We were aware of such cases, but initially did not include the content due to the small number of cases for which we were able to confirm details. But your suggestion made me realize that this manuscript should include this content. I have added the discussion to the last of “paragraph4: COVID-19 and FT1D”. If there are any improvements, please let me know and we will respond immediately.
- The following publication need to be discussed and cited Shirakawa J. Pancreatic β-cell fate in subjects with Covid-19. J Diabetes Investig. 2021;12:2126–2128. doi: 10.1111/jdi.13671
→Thank you. I consider the review you provided to be important for this manuscript. This review argues that SARS-COV-2 causes pancreatic β-cell damage and pancreatic β-cell trans-differentiation. We also had discussed this briefly, using the same references cited in this review. It was written in the first half of the paragraph; COVID-19 and FT1D. Specifically, we speculate that quantitative and functional decline in pancreatic β-cells hastens the course of all types of diabetes. Another notable aspect of this review, candidate viral receptors NRP1 other than ACE are presented. We added this new candidate to the manuscript and included this review as a reference
- Clarify more the possible role of islet autoantibodies, genetic background (HLA), and their relationships in the development of FT1D.→Thank you for your suggestion. Certainly, in the "paragraph 5:acquired immunity", the significance of GAD antibodies is written, but HLA is omitted. Regarding HLA, it is written in "paragraph 4 etiology", but I will also add simple sentence about HLA in "paragraph 5:acquired immunity" so that readers can recall the contents of paragraph 4.  In addition, the fact that DRB1*04:05-DQB1*04:01 and DRB1*09:01-DQB1*03:03 were associated not only with AT1D but also FT1D raises debate about involvement of autoimmunity in FT1D.
- Several data demonstrated increased ACE2 expression in pancreas. This may favor increased pancreatic cells binding of SARS-CoV-2 and hence islet damage. Please discuss this idea and how it may lead to FT1D in COVID-19. Discuss the possible direct infection of the pancrease by the virus (use these references SARS-CoV-2 infects human pancreatic β cells and elicits β cell impairment. Cell metabolism, 2021: p. S1550-4131(21)00230-8. pmid:34081912 and Tang X., et al., SARS-CoV-2 infection induces beta cell transdifferentiation. Cell metabolism, 2021: p. S1550-4131(21)00232-1. pmid:34081913)
→Thank you. Your suggestion is also an interesting problem for me. Even Japan has no reports of COVID-19-associated FT1D despite the prevalence of COVID-19. Personally, I believe that the confirmed SARS-COV2-induced pancreatic β-cell damage and macrophage activation meet the requirements for SARS-COV-2 to develop FT1D.But unexpectedly, there are still no such case reports. Therefore, I thought it would be confusing to discuss the mechanism by which SARS-COV-2 causes FT1D.The two references you have presented are adopted and discussed in the original manuscript. We added that SARS-COV-2 activates macrophages and argued that it is unclear whether this phenomenon is associated with diabetes mellitus.
- Discuss any possible association between race/ ethnicity and COVID-19 induced FT1D. Association between COVID-19 and T1D differ by race, with the largest risks being among American Indian/Alaskan Natives followed by Asian/Pacific Islanders, Black, Hispanic, and White patients. This may reflect differential risk trajectories of T1D by race/ethnicity that preexisted even before the pandemic. For example, one study found that between 2011 and 2015, the incidence of T1D increased among NH Black patients by 4% per year (95% CI: 1.7, 6.3), Hispanics by 2.5% per year 95% CI: 0.5, 4.6), and Asian/Pacific Islander patients by 8.5% per year (95% CI: 3.2, 14.0), while NH White patients did not see significant annual increases (0.5, 95% CI: -0.7, 1.7) (Trends in incidence of type 1 and type 2 diabetes among youths—selected counties and Indian reservations, United States, 2002–2015. Morbidity and Mortality Weekly Report, 2020. 69(6): p. 161. pmid:32053581 ) Even though the reasons for these differences by race/ethnicity remain unclear. Also, discuss age. Increased risk of COVID-19-related T1D among older adults, while T1D usually affects the younger population (The associations between COVID-19 diagnosis, type 1 diabetes, and the risk of diabetic ketoacidosis: A nationwide cohort from the US using the Cerner Real-World Data https://doi.org/10.1371/journal.pone.0266809)→Whether and how FT1D is induced by COVID-19 are important questions. However, there are no case report at this time. I personally expect that if FT1D and COVID-19 are related, we will see higher frequencies in susceptible East Asia. However, COVID-19 may also transcend racial differences and affect different races, like ICI-related T1D. Since there are no case reports at this time and cannot be cited, please let me omit this discussion.
As pointed out, the fact that the impact of COVID-19 on type 1 diabetes incidence varies by race and age was also missing. However, this discussion is complicated, so I added it in simple sentences as much as possible. “Recent real-world data indicates COVID-19 diagnosis is associated with increased risk of new-onset T1D in American Indian/Alaskan Native, Asian/Pacific Islander, Black populations, infants and the elderly.”
- Proof read and check the language of the manuscript. For example:
Line 80 spell out DKA= diabetic ketoacidosis (DKA) →Confirmed and fixed
Line 81 add “is”= in both is drastically different →Confirmed and fixed
Line 106= replace which by and →Confirmed and fixed
Line 108, replace its by their →Confirmed and fixed
Line 139= remove ‘the’ and add ‘disease’= at disease onset →Confirmed and fixed

Reviewer 4 Report (New Reviewer)
The authors review the clinical characteristics of a rare form of diabetes, fulminant type 1 diabetes (FT1D), in comparison to autoimmune type 1 diabetes. They include the clinical characteristics, prevalence, current treatment strategies, issues with treatments, genes that lead to disease, and potential viruses that may lead to development of FT1D. The authors also discuss diabetes complications in association with the viruses, the possibility of COVID-19 in causing FT1D, and immune mechanisms behind the development of FT1D. Overall, I found this review interesting and relevant. I just have a few comments:
1. Please standardize the use of acronyms. You define T1D, but they are not used consistently throughout the text (i.e. lines 321, 435,453 and others you spell out type 1 diabetes instead of using T1D).
2. Please define acronyms at their first use (i.e. PD-1, NOD).
3. Review formatting and grammar. Some places did not have spaces between words (i.e. lines 289 and 294), font type was not consistent through the text, check line 67 for number format, sometimes indentations were used and others they were not, and spacing was not consistent.
4. Change the discussion from second person to first or third person. The journal is applicable to a variety of disciplines, and not all are clinicians. Shifting from second person will make the conclusion more applicable to all readers of the journal.
Author Response
Dear reviewer,
Your meaningful suggestions have changed our manuscript for the better. Thank you.
I have proofread one part of the manuscript so that the reader can read it more easily.
- Clinical Characteristics
To date, most reported FT1D cases have been from East Asia, particularly Japan. FT1D accounts for approximately 20% of acute-onset T1D cases in Japan [3], making it an important subtype for investigation.
I changed “acute-onset T1D cases” to “newly diagnosed T1D with ketosis or ketoacidosis”.
.
We believe this change will make it easier for readers to understand. We appreciate your understanding.
Dear Reviewer4
It was an honor to have your review. Thank you for taking time out of your busy schedule to
read this review. I’ve edited my manuscript according to your pertinent indications.
Answers to each question are written in red. Corrections also are indicated in red in the
proofread manuscript.
- Please standardize the use of acronyms. You define T1D, but they are not used consistently throughout the text (i.e. lines 321, 435,453 and others you spell out type 1 diabetes instead of using T1D).
→Thank you for your careful and kind remarks. Certainly, the description of “type1 diabetes” was not unified. I have properly converted “type1 diabetes” to the “T1D”. The same error also had occurred in “diabetic ketoacidosis” and “DKA”, “pancreatic β-cell” and “β-cell”, so I unified them appropriately.
- Please define acronyms at their first use (i.e. PD-1, NOD).→ Thank you for your pointing out. I reviewed my manuscript carefully and found similar errors in other words and fixed them.
- Review formatting and grammar. Some places did not have spaces between words (i.e. lines 289 and 294), font type was not consistent through the text, check line 67 for number format, sometimes indentations were used and others they were not, and spacing was not consistent.
As you pointed out, L289 and L294 should have been “pancreatic β”, but they were “pancreaticβ”. I fixed it.
Number of patients on line 64ï¼›1,20,000 has been corrected to 120,000. Additionally, I fixed the placement of spaces to separate paragraphs.
- Change the discussion from second person to first or third person. The journal is applicable to a variety of disciplines, and not all are clinicians. Shifting from second person will make the conclusion more applicable to all readers of the journal.
→I changed “If you encounter such cases, it is necessary to ask in detail whether the patient has symptoms of sudden hyperglycemia such as thirst, polydipsia, and polyuria.”
to
“If clinicians encounter such cases, it is necessary to ask in detail whether the patient has symptoms of sudden hyperglycemia such as thirst, polydipsia, and polyuria.”.

Round 2
Reviewer 3 Report (New Reviewer)
The authors have taken my comments into consideration.
One minor comment is to compare between T2D, T1D, FT1D in relation to COVID-19 in Table format.
Also, include the following relevant references:
1- Metabolic Signatures of Type 2 Diabetes Mellitus and Hypertension in COVID-19 Patients With Different Disease Severity https://doi.org/10.3389/fmed.2021.788687
2- Macrophage responses associated with COVID-19: A pharmacological perspective https://doi.org/10.1016/j.ejphar.2020.173547
3-
Author Response
Dear Reviewer,
We think it would be an interesting task to put together a new table of associations between COVID-19 and each type of diabetes (fulminant, type1, and type2). We have spent a few days trying to create a table that will give readers a better understanding. However, at the moment, we decided that we don't have enough evidence to create the table.
At this time, epidemiological studies have shown an increase in type 1 and type 2 diabetes during the COVID-19 pandemic, and this fact and possible mechanisms are included in our review. To create new table, we have to wait for the detailed mechanism and further information to accumulate in the future. In addition, no association between FT1D and COVID-19 has been reported at this time, and even if I create a new table, most of the columns will be filled with "unknown".
Two references were presented as sources for creating the table.
The first reference showed that the increase in COVID-19 disease severity was higher in patients with type 2 diabetes and hypertension, decreased levels of triacylglycerol including certain fatty acids, have been associated with increased severity of COVID-19, and the change of triacylglycerol might be novel predictive markers or therapeutic targets.
The second reference states that SARS-COV-2-induced cytokine storm via macrophage activation is involved in the exacerbation of COVID-19, and that cytokine storm is a therapeutic target for COVID-19.
Both references are interesting, but these present phenomena and biomarkers involved in the severity of COVID-19 rather than diabetes severity.
Therefore, it is also difficult to create a table including these references at the moment, so let it be an important issue in the future. I am sorry that I could not respond to your valuable suggestion.
Notice of change
I have proofread one part of the manuscript so that the reader can read it more easily.
- Clinical Characteristics
To date, most reported FT1D cases have been from East Asia, particularly Japan. FT1D accounts for approximately 20% of acute-onset T1D cases in Japan [3], making it an important subtype for investigation.
I changed “acute-onset T1D cases” to “newly diagnosed T1D with ketosis or ketoacidosis”.
We believe this change will make it easier for readers to understand. We appreciate your understanding.

This manuscript is a resubmission of an earlier submission. The following is a list of the peer review reports and author responses from that submission.
Round 1
Reviewer 1 Report
In their review entitled “Fulminant type 1 diabetes and viruses” Sano and Imagawa give an overview about fulminant type 1 diabetes, that is characterized by insulin-deficiency and severe diabetic ketoacidosis after flue like symptoms and that is in the majority of cases not accompanied by type 1 diabetes associated antibodies. The manuscript is well written and gives a nice overview.
Major points:
- There is some haziness on what is new in this article compared to what has been described in the literature. For example, figure 1 gives an explanation how FT1D could emerge, is this a new concept or who has proposed this? Figure 2 is kind of vague, is it really “only” NK cells that are downregulated/ misfunctioning and leading to islet destruction? The figure would benefit from some more detailed information.
- Could you add one sentence on the incidence/ prevalence of FT1D and AT1D in Japan? This would help to understand the magnitude of the problem.
- The role of viruses and other initiating or accelerating situations are described for their role of FT1D. Viruses are also suspected to contribute to the pathogenesis of type 1 diabetes (AT1D). Please refer to this discussion and outline the major differences here.
- Given the notion that viruses in F1TD play such an important role, please discuss, whether there are any attempts on the way to develop a vaccination programme?
- Section 3.2. now reads “Because these viruses have a cosmopolitan distribution and infect people all over the world, while FT1D occurs in some patients and there have been no reports of mass outbreaks or any region- or season-specific viral effect, it is likely that the interaction of certain viral characteristics and genetic factors, rather than a virus alone, destroys pancreatic β
In addition, because FT1D mainly occurs in adults, the implicated virus is unlikely to affect the host during primary infection. Thus, both viral factors (re-infection by a virus that caused a previous infection but with a different sequence) and host factors (e.g., the impact of aging on viral receptors) are believed to be involved in FT1D”.
I suggest to edit this paragraph, as the first sentence is very long. Probably “cosmopolitan” should be replaced by “global”. Further, the hypothesis that only or mainly adults suffer from FT1D is “because a re-infection seems the cause and therefore the patients need to adult” does not reasonate to me as there are many virus infections and also re-infections before adulthood. In case this is a speculation by the authors, please explain further and add some references. In case this hypothesis has been raised by others, please add that reference.
- In section 4.1. on histopathological findings, please also relate and discuss FT1D to data from npod studies on AT1D.
- Further on section 4.1. it reads: “First, a viral infection of the islets is recognized by RIG-1 and MDA-5, leading to the production of cytokines such as IFNs and chemokines such as CXCL-10, along with infiltration of monocytes and lymphocytes. Simultaneously, macrophages are activated by cytokines and TLRs. The FAS receptor is expressed on pancreatic β cells, promoting apoptosis, and further, nonspecific immune responses destroy pancreatic β cells, α cells, and acinar cells…”
Please clarify, whether this is your own hypothesis or whether you quote here other references/ data?
- Conclusion: Could you give some recommendation for clinical praxis how to detect FT1D early enough? E.g. measurement of ketones and/ or blood glucose in each patients with flu symptoms?
Minor points:
- Please add the reference to the table legends (table 1 and 2) as you mention them in the text.
- Table 1: “Normal ~7ï¼…“ probably should read < 8.7%? Or 6.5% - 8.7%?
- Page 2: “But recently, immune checkpoint inhibitors (ICIs) have been 68 widely used as anti-cancer drug, and several reports suggest that atypical FT1D (ICI-re- 69 lated FT1D) is one of the immune-related adverse events (irAEs), which deserves more 70 attention” – please add a reference here.
- An edit of a native English speaker would help
Author Response
please see the attachement.

Reviewer 2 Report
The article entitled “Fulminant type 1 diabetes and viruses” is an interesting review article describing the relationship between different viral infections and FTD1 pathophysiology. Authors also provide information on potential pathophysiological mechanisms behind this association. This is particularly useful for the clinicians because this form of T1D is relatively rare. The manuscript requires English editing and different clarifications as per the comments attached below:
-Lines 18-19: “from that occurring in autoimmune type 1 diabetes”
-Line 37: “ performed 1–5 months…”
-Line 42: “of the novel severe acute respiratory syndrome coronavirus-2 (SARS-CoV-2) - causing the coronavirus disease 2019 (COVID-19) – on the…”
- Provide references for line 48, line 51, line 52 and line 55
-Line 55: “which develops gradually over several months”
-Line 58: C-peptide
-Line 61: “exocrine pancreatic enzymes”
-Line 62: “pancreatic acinar cells”
-Lines 62-63: “while AT1D is specific to pancreatic β cells.” : this is not true, AT1D involves abnormalities of glucagon-secreting alpha cells and exocrine pancreas (PMID: 26318606).
-Line 67: “this T1D subtype”
-Line 69: “anti-cancer drugs”
-Lines 71-72: please rephrase and improve English, this sentence is not clear enough.
-Line 71: provide a reference for ICI-re-lated FT1D
-Line 75 “Amylase alpha-2A and heat shock protein 10 (HSP10) autoantibodies”
-Line 83: “for a certain period (even decades) after onset; on the other hand….”
-Line 83 “for a certain period (even decades) after onset”: authors should quote here these papers: PMID: 24121625 and PMID: 31646605
-Line 85: insert a period after “diagnosis” and a reference. Then “These factors lead…”
-Lines 92-93: add “and new closed-loop systems”
-Here, discuss more in detail the alteration in glucagon secretion observed in patients with FT1D and quote this paper PMID: 30203124
-Table 1: amend “decline in insulin secretion in AT1D is slower than in FT1D” ; provide a reference for 8.7% as the average of HbA1c level (and for percentages referring to flu-like symptoms too); “DKA in FT1D: abrupt (within approximately 7 days)” ; “Elevation of pancreatic enzyme levels: frequent….rare”; “Anti-GAD antibody: mostly positive in AT1D”; “Susceptible HLA haplotypes in the Japanese population”
-Table 2 caption: specify that these are the Japan Diabetes Society criteria for definite diagnosis of FT1D. Also, provide quotation of the corresponding reference in the reference list. Abbreviate National Glycohemoglobin Standardization Program;; “upper respiratory infection symptoms”; (A) and antibodies to “zinc transporter-8 (ZnT8)”
-Table 2: indicate in the table and table legends the references where all the numbers come from : e.g. HbA1c <8.7%
- Line 100: write in full and then abbreviate “anti-GAD antibodies” the first time you mention it in the text
- -Lines 110-113: please improve English, it is not clear enough
- -Lines 124: weaken taurine’s protective
- Line 128: , overlaps
- -Line 130: , which is…
- -Line 131: is the ligand of integrin subunit beta 7? Please, clarify
- Line 139: replace “with FT1D” with “in FT1D pathophysiology”
- -Line 146: risk factor for FT1D.
- Line 150: replace “destroys” with “leads to destruction of”….
- Line 151: “since” instead of “beacause”
- Line 153: “but with a different sequence” please clarify
- Line 169: skin lesions
- Line 171: of patients
- Line 172: have also been reported
- Line 180: “and the recovery of immune functions, which destroys the islets” : do authors mean the subsequent exuberant immune reponse wthich destroys the pancreatic islets?
- Line 187: virus-induced diabetes
- Line 190: others did not report this association
- Line 193: provide a reference for this statement
- Line 196: “>9% HbA1c”: “or a value of HbA1c at onset <9%”
- Line 197: “evidence of COVID-19-related increased…”
- Line 198: “case of COVID-19-related FT1D..”
- Line 207: is different: SJL/J and…..
- Reference 40 should be inserted in full in the reference list
- Line 211: “exocrine pancreatic region”
- Lines 209-211: improve the structure and punctuation of this sentence
- Line 213: “followed by the rapid disappearance…and by the appearance…”
- Line 219: immunity against viral
- Line 223: EMCV infection
- Line 224: “Experiments using pancreatic T- and B-cell-deficient mice have shown no effects on diabetes,”: do authors mean that T- and B-cell-deficient mice were resistant to diabetes development? Please, clarify
- Line 227: major role in the pathophysiology of fulminant diabetes in the mouse model
- Line 231: Figure caption: specify that these are murine islets.
- Line 240: ref 25 refers to a case report on a different topic, please clarify
- Line 247: NPF [27]: ref 27 refers to a different topic, please clarify
- Paragraph 3.7: cite and briefly discuss data of this paper : PMID: 32675150
- Lines 283-285: “autoptic pancreas” (even line 289 and 317) ; human pancreata? From how many subjects? subjects with a clinical diagnosis of FT1D shortly after the onset? Please clarify these important aspects
- Line 300: further nonspecific
- Line 310: “the decrease in and failure of NK cell activation” decrease in NK cell number and activation?
- Line 311: the infected virus: the infected islet cells?
- Line 330: Non-Obese Diabetic (NOD) mice
- Line 332: as it occurs in FT1D
- Line 335: “as well as the” replace with “along with the”
- Improve resolution of Figure 2
- Line 341: role of viruses in FT1D pathophysiology
- Line 342: “Since” instead of “because”
- Line 343: are aware of it and do not overlook
- Line 345: “following SARS-CoV-2 infection”
- Line 346: it may be plausible that….
- Lines 347-348: improve English
